# A suboptimal OCT4-SOX2 binding site facilitates the naïve-state specific function of a *Klf4* enhancer

**Jack B. Waite**[1], **RuthMabel Boytz**[2], **Alexis R. Traeger**[1], **Torrey M. Lind**[1], **Koya Lumbao-Conradson**[1], **Sharon E. Torigoe**[1,2]*

**1** Biochemistry & Molecular Biology Program, Lewis & Clark College, Portland, Oregon, United States of America, **2** Biology Department, Lewis & Clark College, Portland, Oregon, United States of America

* torigoe@lclark.edu

**Data Availability Statement:** All relevant data are within the manuscript and its Supporting Information files. All original gel images are

## Abstract

Enhancers have critical functions in the precise, spatiotemporal control of transcription during development. It is thought that enhancer grammar, or the characteristics and arrangements of transcription factor binding sites, underlie the specific functions of developmental enhancers. In this study, we sought to identify grammatical constraints that direct enhancer activity in the naïve state of pluripotency, focusing on the enhancers for the naïve-state specific gene, *Klf4*. Using a combination of biochemical tests, reporter assays, and endogenous mutations in mouse embryonic stem cells, we have studied the binding sites for the transcription factors OCT4 and SOX2. We have found that the three *Klf4* enhancers contain suboptimal OCT4-SOX2 composite binding sites. Substitution with a high-affinity OCT4-SOX2 binding site in *Klf4* enhancer E2 rescued enhancer function and *Klf4* expression upon loss of the ESRRB and STAT3 binding sites. We also observed that the low-affinity of the OCT4-SOX2 binding site is crucial to drive the naïve-state specific activities of *Klf4* enhancer E2. Altogether, our work suggests that the affinity of OCT4-SOX2 binding sites could facilitate enhancer functions in specific states of pluripotency.

## Introduction

Enhancers have critical roles in the spatiotemporal regulation of transcription during development [1–3]. These *cis*-regulatory elements modulate transcriptional activation through the recruitment of transcription factors (TFs) to their cognate binding sites, facilitating subsequent interactions in the assembly of enhanceosome complexes and the transcription machinery. Many mutations in enhancers have been attributed to diseases and evolution [1, 4–8], underscoring the importance of the sequences themselves in directing enhancer functions. Hence, a deeper insight into the code of enhancers will enable us to better decipher how genomic information controls transcription during development.

Enhancer grammar has emerged as one model towards understanding how transcription regulation is encoded into enhancer sequences. This viewpoint proposes that constraints on TF binding sites (TFBSs), such as their type, number, binding affinities, and relative

uploaded on the Open Science Framework (https://doi.org/10.17605/OSF.IO/S2KQY).

**Funding:** This work was supported by Start-Up funds from Lewis & Clark College, the John S. Rogers Summer Science Program, and grants to S.E.T. from the M.J. Murdock Charitable Trust (Awards 2016190 and NS-2019136792), the Medical Research Foundation of Oregon (New Investigator Grant, 2022-2024), and the National Science Foundation (Award 2117304). The funders had no role in study design, data collection and analysis, decision to publish, or preparation of the manuscript.

**Competing interests:** The authors have declared that no competing interests exist.

arrangements, control the frequency and stability of TF-DNA and TF-TF interactions at enhancers [9–12]. While TFBS type has long been appreciated in specifying enhancer function, there is a growing body of evidence to demonstrate the importance of grammatical features like binding affinity and syntax [8, 11, 13]. However, we have only just begun to elucidate the grammatical constraints for developmental enhancers, restricting our ability to predict functions from enhancer sequences.

Pluripotent stem cells (PSCs) represent the earliest stages of development. In mammals, several states of pluripotency have been defined and characterized: naïve state, formative state, and primed state [14–16]. The naïve and primed states correspond to pre- and post-implantation embryos, respectively, and the formative state is considered an intermediate stage in between the naïve and primed states (S1 Fig). While all three states share some commonalities in their transcriptional profiles, such as the expression of the core pluripotency TFs OCT4, SOX2, and NANOG, there are distinct differences [17–20], raising questions about how transcription is differentially regulated in each pluripotency state.

There have been several studies of enhancer grammar in pluripotency, which have highlighted the importance of heterotypic clusters of TF binding sites [21, 22] and identified some constraints in syntax between TFBSs [23, 24]. However, while these studies focused on enhancers active in naïve-state mouse embryonic stem cells (mESCs), it remains unknown whether these findings apply to naïve-state-specific enhancers or to enhancers that are active across multiple states of pluripotency. To address this gap in knowledge of transcription regulation in pluripotency, we sought to identify enhancer grammar constraints that mediate gene expression differences between the states of pluripotency.

To investigate whether there are naïve-state-specific grammatical constraints for enhancers, we focused on the enhancers for the *Klf4* gene. Best known as one of the Yamanaka factors for inducing pluripotency [25, 26], *Klf4* is expressed in the naïve state of pluripotency, but not the formative or primed states [27, 28]. KLF4 is a key TF in the establishment and maintenance of the naïve-state expression profile [27, 29, 30]. Three downstream enhancers for *Klf4* involve the hierarchical and cooperative recruitment of several pluripotency TFs: OCT4, SOX2, ESRRB, and STAT3 (S2 Fig). Of these TFs, OCT4 and SOX2 are the lead factors for enhanceosome assembly [28]. Given their central role in *Klf4* transcription regulation, we asked whether there are grammatical constraints for the OCT4 and SOX2 binding sites in the *Klf4* enhancers and, if so, whether those underlie the naïve-state-specific function of these enhancers.

## Results

### The *Klf4* enhancers contain low-affinity binding sites for OCT4 and SOX2

OCT4 and SOX2 binding sites are often observed immediately adjacent to each other, forming a composite oct-sox motif [31, 32] (Fig 1). There are notable, functional alternatives to this arrangement of binding sites, such as a spacing or three base pairs in the *Fgf4* enhancer [33] and a compressed motif for OCT4 and SOX17 [34]. Nonetheless, immediately adjacent binding sites is most optimal for cooperative DNA binding by OCT4 and SOX2 [35]. While there is some flexibility for spacing, the orientations of the individual binding sites are constrained to mediate cooperative binding [35]. The optimal spacing and orientation is observed in a consensus motif for the OCT4-SOX2 composite site and the OCT4-SOX2 binding sites in the three *Klf4* enhancers (Fig 1). Prior work has shown that this arrangement is critical for their synergistic function in the transcription activation of *Klf4* [28]. Thus, it appeared unlikely that the spacing and orientation of the OCT4 and SOX2 binding sites contribute to the naïve-state specific function of the *Klf4* enhancers.

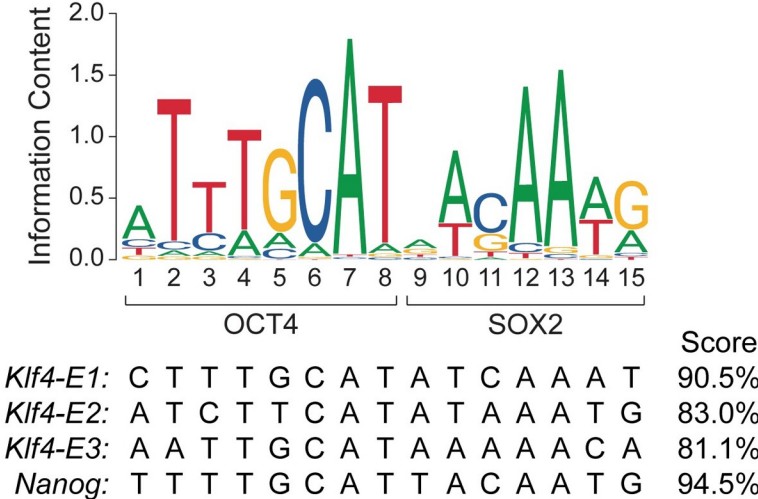

**Fig 1. Sequence analysis predicts suboptimal function for the OCT4-SOX2 composite sites in the *Klf4* enhancers.** (upper) A sequence logo diagram of a position frequency matrix for the OCT4-SOX2 composite binding site [36]. (lower) Alignment of the OCT4-SOX2 composite binding sites from the *Nanog* enhancer and the three *Klf4* enhancers. The score is the relative score, as calculated on the JASPAR database [36].

However, there is growing evidence that low-affinity TF binding sites are key to the spatio-temporal specificity of enhancers [8, 11, 13], so we next assessed the binding affinity of the OCT4 and SOX2 binding sites in the *Klf4* enhancers. We scored the composite OCT4-SOX2 binding site (OS site) sequences from the *Klf4* enhancers E1, E2, and E3 by using a position frequency matrix (PFM) for a composite OS site in the JASPAR database [36]. For comparison, we also analyzed the OS site sequence from the *Nanog* enhancer [37, 38], which previously has been used in biochemical experiments because of its high affinity [28, 39]. The *Nanog* OS site showed excellent alignment with the PFM for the composite OS site and had a high relative score of approximately 95% (Fig 1). However, the OS sites in the *Klf4* enhancers had relative scores in the range of 81% to 91% (Fig 1), which suggests lower binding affinities. Notably, the relative scores for the OS sites from enhancers E2 and E3 were just above 80%, which is the typical threshold for the prediction of putative TF binding sites on the JASPAR database.

To validate our sequence analysis, we assessed the binding affinities of the OS sites from the *Klf4* and *Nanog* enhancers by electrophoretic mobility shift assays (EMSAs) with purified OCT4 and SOX2 proteins. Whereas the *Nanog* OS site exhibited high occupancy with the OCT4-SOX2 heterodimer complex, all three OS sites from the *Klf4* enhancers displayed a significantly lower amount of binding for the OCT4-SOX2 heterodimer complex under the same binding conditions (Fig 2A and 2B). While the low level of unbound DNA indicates high occupancy for the OS site from the enhancer E1, the majority of the shifted DNA molecules is bound only by OCT4, not the OCT4-SOX2 heterodimer. To achieve the same amount of binding by the OCT4-SOX2 heterodimer as the *Nanog* OS site, 4–8 times more protein was required for the OS site from enhancer E1 (S3A and S3B Fig). For the OS sites from enhancers E2 and E3, there was substantially reduced occupancy of DNA compared to the OS site from *Nanog*. For the OS site from enhancer E2, comparable occupancy with the OCT4-SOX2 heterodimer to the *Nanog* OS site was achieved with about 4 times more protein (S3C and S3D Fig). Even with 32 times more protein, we could not observe occupancy of the OS site from enhancer E3 with the OCT4-SOX2 heterodimer at the same level as the *Nanog* OS site (S3E

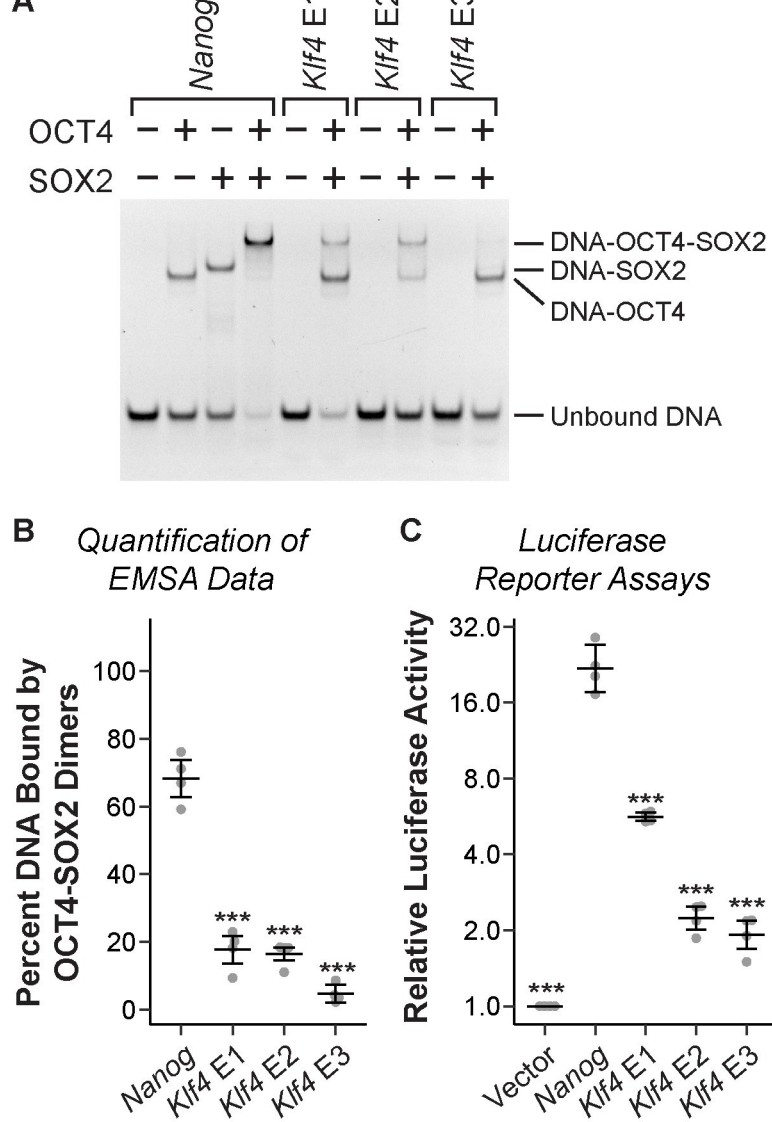

**Fig 2. The OCT4-SOX2 composite sites from the *Klf4* enhancers exhibit suboptimal binding affinities and activities.** (A) Electrophoretic mobility shift assays (EMSAs) were performed using Cy5-labeled DNA probes, containing a single OCT4-SOX2 composite site (OS site) from the *Nanog* or *Klf4* enhancers. Binding reactions contained 2 nM of Cy5-labeled DNA probe, 19 nM OCT4 and/or 16 nM SOX2. Products were separated by nondenaturing gel electrophoresis, and gels were imaged to detect the Cy5 label. The + and –symbols indicate presence or absence of the indicated protein. Three replicate experiments were performed, and a representative image is shown. (B) Quantitative analysis of EMSA data to measure OCT4-SOX2 binding to OS sites from the *Nanog* and *Klf4* enhancers. The % DNA bound by OCT4-SOX2 dimers (amount of signal for the DNA-OCT4-SOX2 band/total DNA signal in the lane) is shown. Error bars represent mean of replicates ± 95% confidence interval. *Asterisks* represent statistically significant differences compared to the *Nanog* OS site (***$p < 0.001$, unpaired t-test). (C) A single copy of the OS sites from the *Nanog* or *Klf4* enhancers was inserted upstream of a minimal promoter to control expression of firefly luciferase. Constructs were transfected into mESCs for dual-luciferase reporter assays. Firefly data was normalized to expression of the *Renilla* luciferase and then to empty vector. Error bars represent mean of biological replicates ± 95% confidence interval. *Asterisks* represent statistically significant differences compared to the *Nanog* OS site (***$p < 0.001$, unpaired t-test).

and S3F Fig). Thus, our EMSA results show that the OS sites in the *Klf4* enhancers are suboptimal for binding affinity.

To evaluate whether these reduced binding affinities affect transcriptional activities, we performed luciferase reporter gene assays in mESCs. Our reporter constructs each contained a single copy of an OS site that was inserted upstream of a minimal promoter. Results from these reporter assays were consistent with our observations for binding affinity (Fig 2C). The high-affinity OS site from the *Nanog* enhancer led to very high levels of luciferase expression, at about 20-fold above empty vector. The OS site from enhancer E1 had a relatively modest effect, about 5-fold above empty vector. Notably, the OS sites from enhancers E2 and E3 only minimally increased luciferase expression, about 2-fold above empty vector, which is consistent with our EMSA data.

## The low-affinity OCT4-SOX2 site underlies naïve-state specific activity of *Klf4* enhancer E2

We were intrigued that the OS sites in all three *Klf4* enhancers are suboptimal for OCT4 and SOX2 binding, especially as OCT4 and SOX2 are the lead factors for enhanceosome assembly at these enhancers [28]. The low-affinity of the OS site in *Klf4* enhancer E2 was particularly notable, as deletion of this site alone was sufficient to disrupt ~85% of *Klf4* expression in mESCs [28]. Thus, we sought to understand the functional role for the low affinity of the OS site in *Klf4* enhancer E2.

It has been proposed that enhancers containing low-affinity TF binding sites are only active when there are sufficiently high concentrations of the respective TFs, limiting their spatiotemporal functions [11, 13, 40]. This is an appealing model to explain the naïve-state specificity of *Klf4* enhancer E2. While OCT4 levels do not appreciably change from naïve to primed state, some studies have reported that SOX2 levels decrease as cells exit the naïve state [18, 41]. Hence, when SOX2 levels decrease, its concentration may be too low to occupy the SOX2 binding site in *Klf4* enhancer E2 sufficiently, leading to loss of expression. However, we wondered whether this explanation might be incomplete, given that *Klf4* expression is also regulated by ESRRB and STAT3, which are recruited by OCT4 and SOX2 to the enhancers [28]. Whereas OCT4 and SOX2 are utilized across all states of pluripotency, ESRRB and STAT3 are considered hallmarks of the naïve state and can facilitate conversion of primed-state PSCs to the naive-state [42, 43]. Notably, loss of either the ESRRB site or the STAT3 site leads to a 50% decrease in *Klf4* expression [28].

Thus, we postulated that the low-affinity of the OS site drives the requirement of ESRRB and STAT3 contributions to achieve wild-type levels of enhancer function in the naïve state. Conversely, substitution with a high-affinity OS site might be sufficient to achieve wild-type levels of enhancer activity in naïve-state PSCs, even in the absence of ESRRB and STAT3. To test our proposed mechanism, we generated three mutations in *Klf4* enhancer E2: (1) replacement of the low-affinity OS site with a high-affinity sequence (OS+); (2) loss of the ESRRB and STAT3 sites (ΔES); and (3) combination of the high-affinity OS site substitution with loss of the ESRRB and STAT3 sites (OS+/ΔES) (Fig 3A).

We first assessed the impact of these mutations on enhancer activity in the naïve state using dual luciferase reporter assays in naïve-state mESCs. For our luciferase reporter assay constructs, we inserted the *Klf4* enhancer E2 sequence about 50 bp upstream of a minimal promoter (S4 Fig). The OS+ mutant showed a 60 times more enhancer activity compared to the wild type enhancer (Fig 3B), which is consistent with our finding that the OS site in *Klf4* enhancer E2 is suboptimal (Fig 2). By replacing the wild-type, low-affinity OS site with a high-affinity version, OCT4 and SOX2 bind to the OS+ mutant much more often and, subsequently,

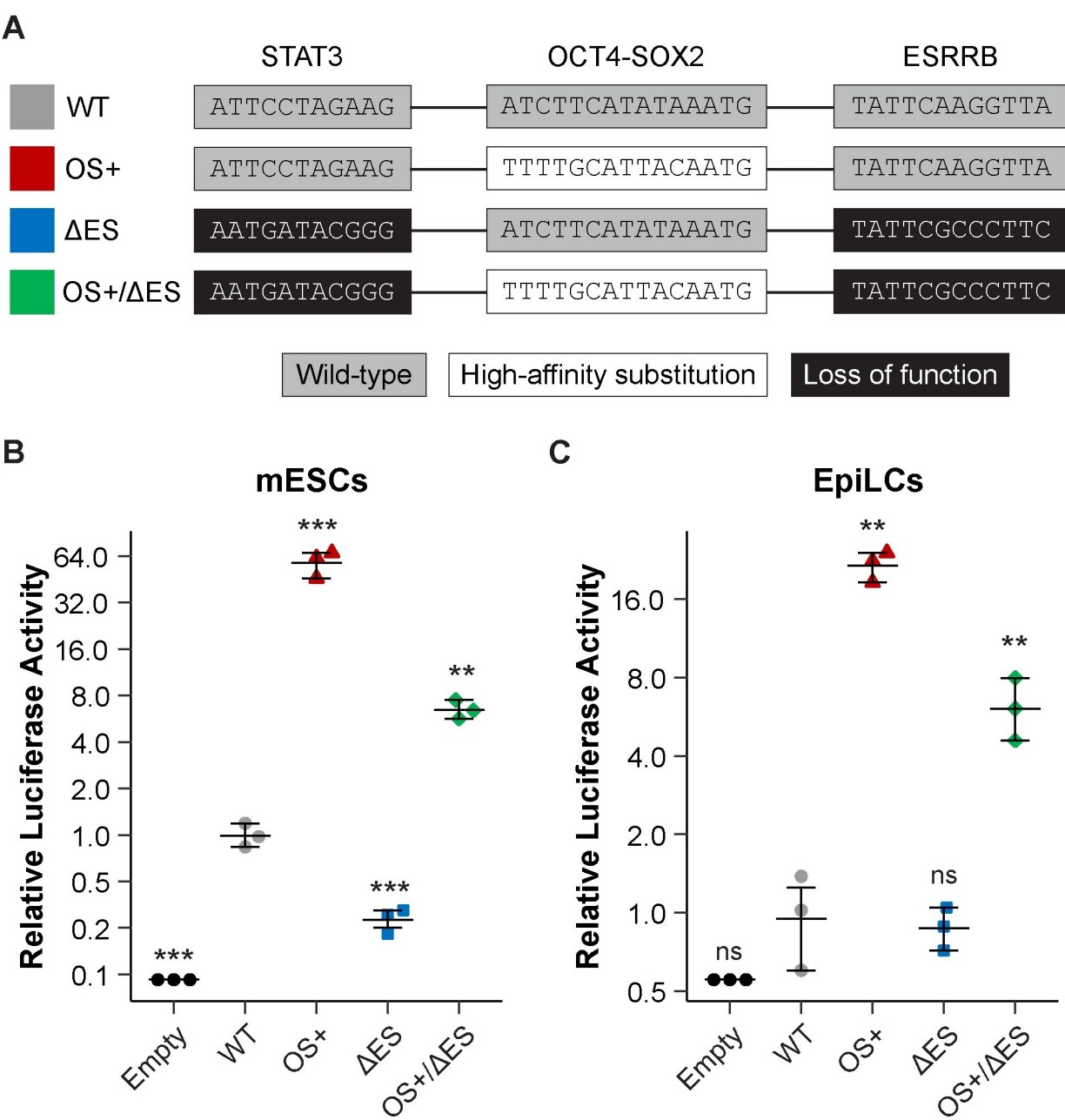

**Fig 3. A low-affinity OCT4-SOX2 motif facilitates naïve-state specific activity of the *Klf4* enhancer E2.** (A) Schematic representations of the transcription factor binding sites in the wild-type (WT) and mutant *Klf4* enhancer E2 sequences. (B, C) The WT and the indicated mutant enhancer E2 sequences were assessed by dual luciferase reporter assays in naïve-state mESCs (B) or formative-state EpiLCs (C), which were generated by growing mESCs in EpiLC induction media. Firefly data was normalized to *Renilla* and then to the empty vector. Error bars represent mean of biological replicates ± 95% confidence interval. *Asterisks* represent statistically significant differences compared to WT (**p<0.01; ***p<0.001, unpaired t-test). ns (not significant).

facilitate more recruitment of other TFs, like ESRRB and STAT3. Hence, the OS+ mutant leads to a dramatic increase in transcription activation. By contrast, the ΔES mutant, in which we have sought to disrupt the ESRRB and STAT3 binding sites, exhibited an approximately 75% loss in activity compared to the wild-type enhancer (Fig 3B). Upon loss of the ESRRB and STAT3 binding sites, these additional transcription factors cannot bind to the enhancer. Thus,

the ΔES mutant is primarily reliant on the binding of OCT4 and SOX2 to a low-affinity sequence, leading to relatively low levels of transcription activation. This reiterates that ESRRB and STAT3 have important contributions to enhancer E2 function [28].

Importantly, the OS+/ΔES mutant exhibited a 6.5 times greater enhancer activity than wild-type (Fig 3B). This is substantially lower compared to the OS+ mutant, further supporting that ESRRB and STAT3 have significant contributions to enhancer E2 activity. Nevertheless, the OS+/ΔES mutant is more active than the wild-type enhancer E2. Hence, the high-affinity OS site can compensate for the loss of ESRRB and STAT3 sites, resulting in at least the same level of activity as the wild-type enhancer in naïve-state mESCs. These data indicate that, in the context of a low-affinity OS site, ESRRB and STAT3 are required for *Klf4* enhancer E2 function, but a high-affinity OS site is sufficient to facilitate at least wild-type levels of enhancer function, independent of ESRRB and STAT3 binding sites.

We then asked whether a high-affinity OS site might render *Klf4* enhancer E2 active as cells exit the naïve state and transition towards the primed state. Based on our proposed model above, the low-affinity OS site is crucial to support the naïve-state specific activity of *Klf4* enhancer E2. When ESRRB and STAT3 functions are lost in the transition from naïve state to primed state, the low-affinity OS site is insufficient to drive *Klf4* enhancer E2 activity and subsequent activation of *Klf4* expression. Thus, the OS+ and/or OS+/ΔES mutants might exhibit aberrant transcriptional activity in formative- or primed-state PSCs.

To address this question, we performed dual luciferase reporter assays on our *Klf4* enhancer E2 mutations in epiblast-like cells (EpiLCs), which are generated from naïve-state mESCs and mimic formative-state PSCs [16, 44]. Importantly, in EpiLCs, *Klf4* and *Esrrb* are significantly downregulated, and STAT3 activities are lost due to withdrawal of its activator LIF from the growth media [44]. We found that a high-affinity OS site in *Klf4* enhancer E2 led to new, aberrant enhancer activity in EpiLCs (Fig 3C). The wild-type *Klf4* enhancer E2 showed very little activity above empty vector, which is consistent with previous characterization that this enhancer is naïve-state specific [28]. Loss of the ESRRB and STAT3 binding sites in the ΔES mutant did not significantly affect enhancer activity, as expected since ESRRB and STAT3 functions are absent in EpiLCs. Notably, the OS+ and OS+/ΔES mutant displayed 25 and 6 times greater activity than wild-type in EpiLCs, respectively, indicating that a high-affinity OS site was sufficient to promote *Klf4* enhancer E2 activity after cells had exited the naïve-state.

It is curious that the OS+/ΔES mutant displayed lower activity than OS+ in EpiLCs (Fig 3C), which indicates that our disruptions to ESRRB and STAT3 had an effect in the context of the high-affinity OS site substitution, despite the lack of ESRRB and STAT3 function in EpiLCs. It is plausible that there might be very low levels of ESRRB remaining in EpiLCs. As OCT4 and SOX2 are the lead TFs that facilitate subsequent recruitment of ESRRB and STAT3 [28], the high-affinity OS site may increase OCT4 and SOX2 occupancy at the enhancer and, thus, promote more ESRRB binding to elevate transcriptional activation.

Altogether, our analysis of *Klf4* enhancer E2 mutants by luciferase reporter assays in naïve-state mESCs and formative-state EpiLCs demonstrate the importance of a low-affinity OS site in mediating the naïve-state specific activity of this enhancer. With a low-affinity OS site, this enhancer requires ESRRB and STAT3 binding sites to function in the naïve-state (Fig 3B) and, thus, has little activity when PSCs exit the naïve state (Fig 3C). However, with a high-affinity OS site, this enhancer no longer requires ESRRB and STAT3 to generate wild-type levels of transcriptional activation in naïve-state mESCs (Fig 3B) and exhibits aberrant activities in formative-state EpiLCs (Fig 3C).

## A high-affinity OCT4-SOX2 site in enhancer E2 rescues *Klf4* expression in naïve-state mESCs upon loss of the ESRRB and STAT3 binding sites

While informative of enhancer activities, luciferase reporter assays isolate enhancers from their native genomic and nuclear contexts, which likely contribute to their functions. To examine the effects of our mutations endogenously, we employed a recombinase-mediated cassette exchange (RMCE) approach [45–47], which produces isogenic cell lines that vary at the *Klf4* enhancer E2 (Fig 4A). We first inserted heterotypic Flp recognition target (FRT) targets around *Klf4* enhancer E2 and an mCherry expression reporter to facilitate selection of genetically altered cells. In this cell line, we introduced an exchange plasmid, which contained either the wild-type or a mutant *Klf4* enhancer E2 flanked by the same heterotypic FRT sites, and Flp recombinase to induce site-specific recombination. Following RMCE, we selected single cells based on the loss of mCherry expression for clonal outgrowth and confirmed integration of the wild-type or mutant *Klf4* enhancer E2 by sequencing. We analyzed the effects of our mutations on gene expression in several different wild type and mutant clones, using quantitative, reverse transcription PCR (RT-qPCR).

*Klf4* mRNA levels in our RMCE-generated cell lines were consistent with our luciferase reporter assays (Fig 4B). In the OS+ cell lines, we observed that *Klf4* expression increased by 75%, which was statistically significant. There also was a 50% loss in *Klf4* expression upon disruption of the ESRRB and STAT3 binding sites (ΔES), which was rescued by the high-affinity OS site substitution (OS+/ΔES). These data reiterate that a high-affinity OS site can overcome the loss of ESRRB and STAT3 binding sites, which are critical for *Klf4* enhancer E2 function. In the OS+, ΔES and OS+/ΔES cell lines, we did not observe significant changes in expression of the adjacent *Rad23b* gene (Fig 4C), emphasizing the specificity of enhancer E2 on *Klf4* expression.

It is notable that the effects of the OS+ and OS+/ΔES mutations on *Klf4* transcription in mESCs (Fig 4B) were considerably lower than the changes observed by reporter assays (Fig 3B). This could be due to the collective functions of multiple enhancers, the promoter and other regulatory factors for *Klf4*. Indeed, when we used the *Klf4* promoter instead of the typical minimal promoter for luciferase reporter assays, we found that the relative effects of *Klf4* enhancer E2 decreased (S5 Fig).

Given the function of KLF4 as a transcription factor, we asked whether changes in *Klf4* expression, due to our enhancer E2 mutations, had effects on the expression of other genes. KLF4 is among the network of pluripotency TFs, which positively regulate each other [31, 48–52], so we measured expression levels for the pluripotency genes *Pou5f1*, *Sox2*, and *Nanog* in our RMCE-generated cell lines. Indeed, we observed modest upregulation in the OS+ cell lines, though it was only statistically significant for *Pou5f1* (Fig 4D–4F). We note that these modest rises in expression could partially explain the elevated *Klf4* mRNA levels, especially since OCT4 and SOX2 are key factors to activate *Klf4* transcription. For the ΔES and OS+/ΔES cell lines, there were no significant changes in expression of *Pou5f1*, *Sox2*, and *Nanog* (Fig 4D–4F). While somewhat unexpected for the ΔES cell lines, previous studies also have reported that loss of *Klf4* does not impact the expression of these genes [28, 53].

As KLF4 is also a significant factor for the establishment and maintenance of the naïve state [27, 29, 30], we also assessed whether our RMCE-generated cell lines displayed effects on the expression of naïve-state specific genes. *Esrrb* and *Tbx3* are well known markers for the naïve state [42, 43, 54], and *Klf2* and *Klf5* are also expressed in the naïve state but not the primed state [55]. Furthermore, the trio of KLF2, KLF4, and KLF5 work in concert to promote naïve-state pluripotency and self-renewal [27, 30, 56]. For the OS+ cell lines, we did not observe statistically significant changes in expression of the naïve-state genes, though *Esrrb* was somewhat

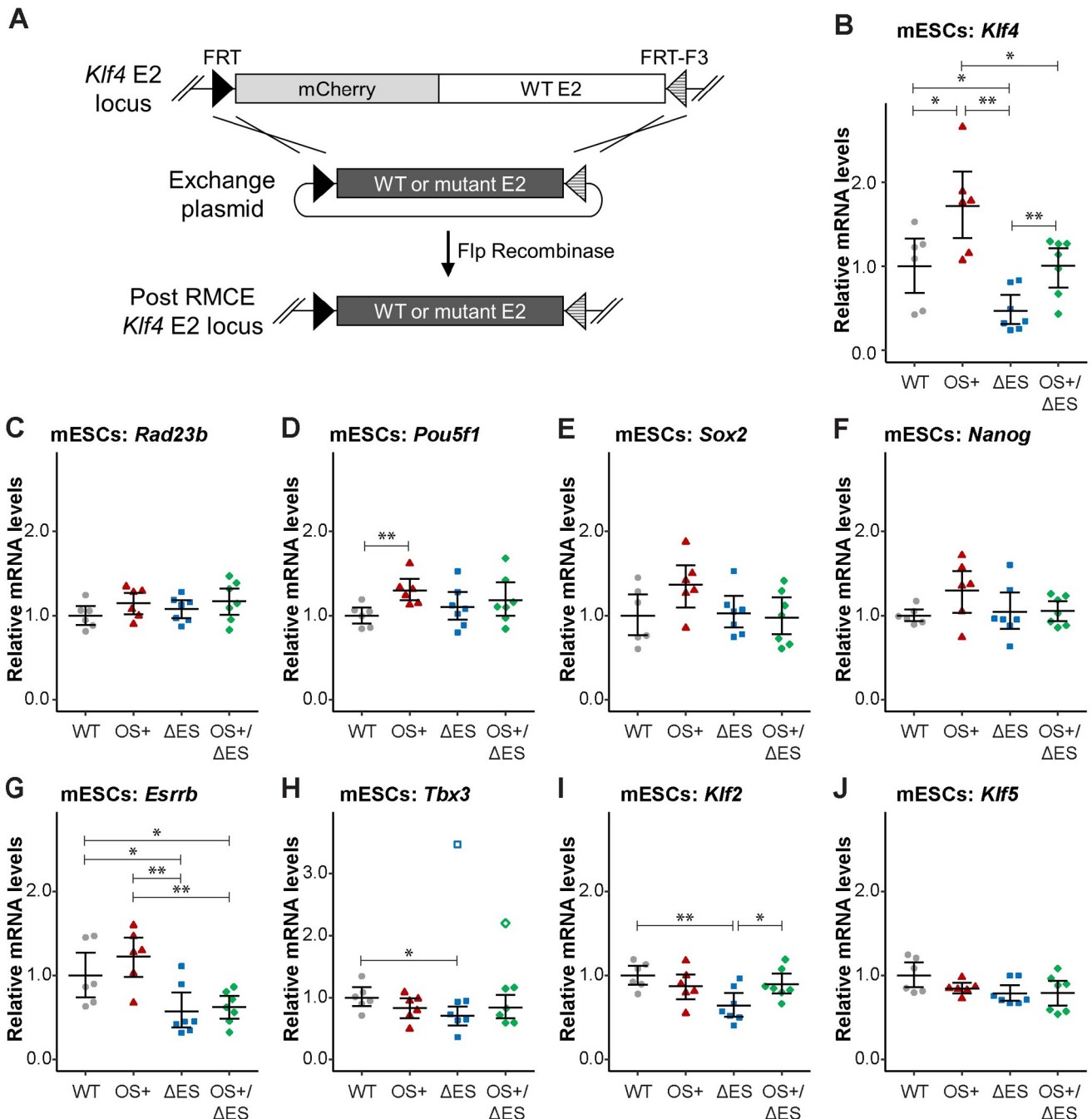

**Fig 4. An endogenous high-affinity OCT4-SOX2 motif restores _Klf4_ expression upon loss of ESRRB and STAT3 sites.** (A) Mutant mESC lines were generated by recombinase-mediated cassette exchange (RMCE). Heterotypic Flp recognition target (FRT) sites and an mCherry expression reporter were inserted around _Klf4_ enhancer E2 in mESCs. In the presence of an exchange plasmid, which contains the enhancer E2 flanked by the heterotypic FRT sites, Flp recombinase facilitates integration of the desired enhancer E2 sequence. Single cells were selected after RMCE, isolated for clonal outgrowth, and confirmed by PCR and sequencing. (B-F) Analysis of gene expression in the mutant mESC lines. For each enhancer E2 sequence, multiple clonal populations were grown under naïve-state conditions, and total mRNA was harvested from each and reverse transcribed into cDNA. Quantitative PCR was performed on the cDNA to measure levels of _Klf4_ (B), _Rad23b_ (C), _Pou5f1_ (D), _Sox2_ (E), _Nanog_ (F), _Esrrb_ (G), _Tbx3_ (H), _Klf2_ (I), and _Klf5_ (J). Data was analyzed using the ΔΔCT method. Each data point represents the average from 2–3 technical replicates for each isolated clone, and error bars represent mean of clones ± 95% confidence interval. _Asterisks_ represent statistically significant differences between averages (*p<0.05; **p<0.01, unpaired t-test), with brackets to indicate samples compared. If not shown, averages were not statistically different. Outliers, as determined by the Grubb's test, are denoted by unfilled shapes and were not utilized in statistical tests.

elevated (Fig 4G–4J). Thus, even though *Klf4* is upregulated in these cells, this may not be enough to have a measureable impact on naïve-state transcription regulation.

While there no change in *Klf5* expression, the ΔES cell lines exhibited a 30–40% downregulation of *Esrrb*, *Tbx3*, and *Klf2* (Fig 4G–4J), demonstrating that the reduced *Klf4* expression, due to the loss of the ESRRB and STAT3 binding sites in enhancer E2, has an impact on some naïve-state transcription regulation. Importantly, while *Esrrb* levels were similar between the ΔES and OS+/ΔES cell lines (Fig 4G), *Tbx3* and *Klf2* expression levels were similar to the isogenic wild-type cell lines (Fig 4H and 4I). These results indicate that rescuing enhancer E2 function with the high-affinity OS site substitution restored *Klf4* expression levels (Fig 4B) and, subsequently, some naïve-state specific gene expression.

Given the observed changes in gene expression, we then asked whether there were broader phenotypic effects on our mutant mESCs, particularly the ΔES cell lines. Colony morphology across all RMCE-generated cell lines was similar, with tightly packed cells into colonies with generally well-defined, smooth edges (S6 Fig). There were some elongated, though not flattened, cells at the periphery of colonies, but this was observed across all cell lines, whether containing the wild-type or a mutant enhancer E2. To assess for cell proliferation potential, we measured the growth rates of our mutant cell lines. We found no significant differences in doubling times between the wild-type and mutant cell lines (S7 Fig). The lack of effects on colony morphology and growth rate is particularly notable for the ΔES cell lines, given downregulation of several naïve-state specific genes (Fig 4G–4I). However, these observations are consistent with previously published work that *Klf4* is largely dispensable for maintaining pluripotency [28, 53, 56].

## Introduction of a high-affinity OCT4-SOX2 site in enhancer E2 does not affect *Klf4* expression in the formative state

Having observed that a high-affinity OS site, in the presence or absence of ESRRB and STAT3 sites, led to *Klf4* enhancer E2 activity in formative-state EpiLCs (Fig 3C), we asked whether endogenous introduction of these mutations might lead to aberrant expression of *Klf4* after exiting the naïve state. We generated EpiLCs from the RMCE-generated wild-type, OS+ and OS+/ΔES mutant cell lines (from Fig 4) that displayed the highest *Klf4* expression in the naïve state. We harvested the EpiLCs to measure mRNA levels by RT-qPCR after two days of induction, when there is significant loss of *Klf4* and *Esrrb* expression and upregulation of *Fgf5* [49] (S8A Fig).

Induction to the formative state was successful, as evidenced by downregulation of the naïve-state genes *Klf4* and *Esrrb* (Fig 5A and 5B) and of *Nanog* and *Sox2* (Fig 5C and 5D). There was a modest decrease in *Pou5f1* expression, though it was generally statistically insignificant (Fig 5E), and there was a marked upregulation of *Fgf5*, a marker for the formative and primed states (Fig 5F). These changes in gene expression are similar to those observed during EpiLC inductions for unedited mESCs that lack the FRT sites at *Klf4* enhancer E2 (S8A Fig), and relative mRNA levels were generally comparable between the unedited cells and our RMCE-generated cell lines (S8B–S8G Fig).

Importantly, across our RMCE-generated wild-type, OS+ and OS+/ΔES cell lines, there was no statistical difference in expression of *Klf4* (Fig 5A) or any of the genes assessed (Fig 5B–5F) in EpiLCs. Thus, while the high-affinity OS site substitution led to increased enhancer E2 activity in luciferase reporter assays (Fig 3C), endogenous introduction of those mutations was insufficient to impact *Klf4* expression in EpiLCs (Fig 5A). This discrepancy might suggest that enhancer E2 is not functional, but we suspect that de-activation of *Klf4* transcription in the formative state involves more pathways than this enhancer (see Discussion).

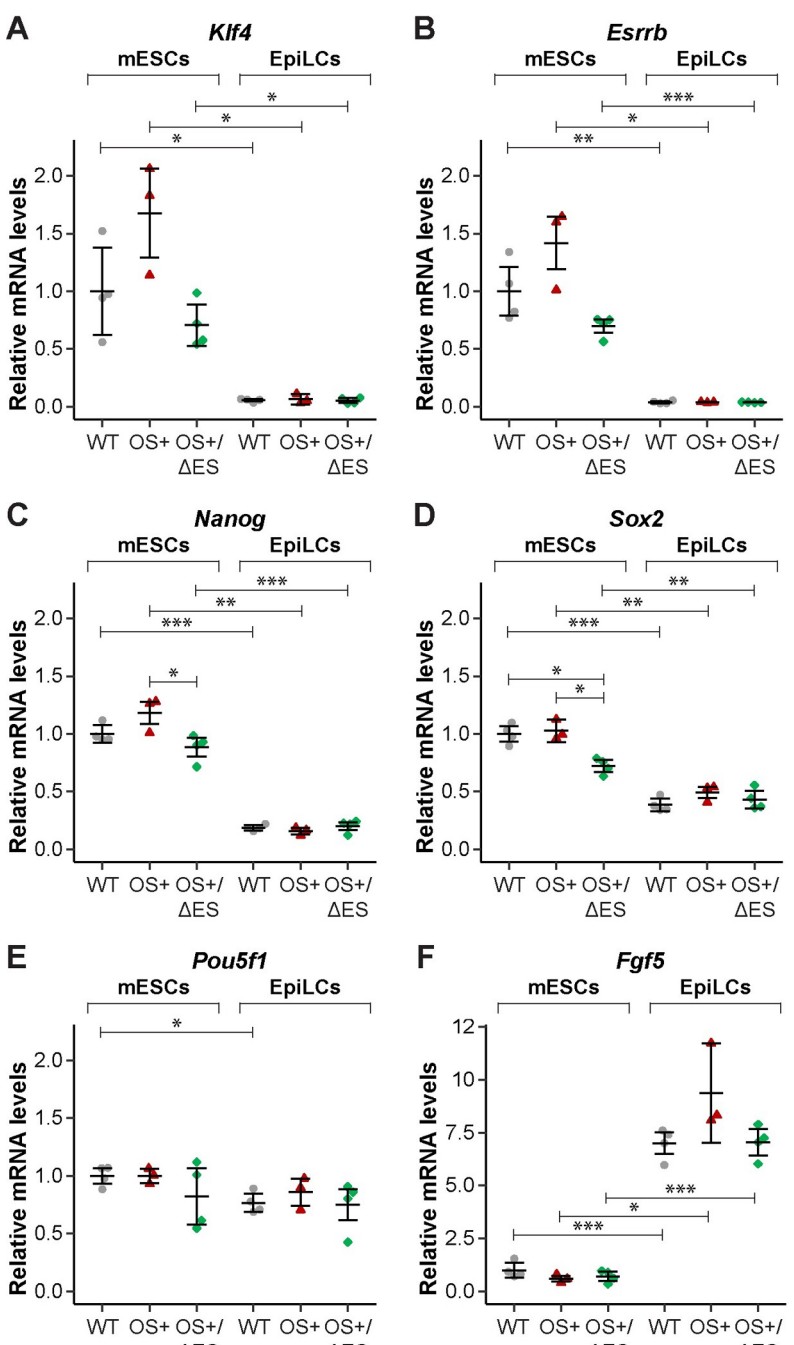

**Fig 5. Increased enhancer E2 function is insufficient to overcome endogenous mechanisms to repress *Klf4* in EpiLCs.** mESCs with endogenous enhancer E2 mutations from Fig 4 were converted to EpiLCs by growth in EpiLC induction media for 48 hours. Total RNA was collected from before and after induction and then reverse transcribed into cDNA. Quantitative PCR was performed on the cDNA to measure levels of *Klf4* (A), *Esrrb* (B), *Nanog* (C), *Sox2* (D), *Pou5f1* (E) and *Fgf5* (F). Data was analyzed using the ΔΔCT method. Each data point represents the average from 2–3 technical replicates for each clone, and error bars represent mean of clones ± 95% confidence interval. *Asterisks* represent statistically significant differences between averages (*p<0.05; **p<0.01, ***p<0.001, unpaired t-test), and brackets indicate samples compared in unpaired t-tests. If not shown, averages were not statistically different.

## Discussion

Here, we report that the three *Klf4* enhancers contain suboptimal binding sites for the lead TFs OCT4 and SOX2. We found that the low affinity of the OCT4-SOX2 site in *Klf4* enhancer E2 facilitates naïve-state specific enhancer function. In the naïve-state, a high-affinity site led to elevated enhancer function and subsequent *Klf4* expression and also overcome the loss of ESRRB and STAT3 binding sites. In the formative state, a high-affinity site resulted in aberrant enhancer E2 activity but could not counteract other repressive mechanisms to de-activate *Klf4* expression.

### Regulation of *Klf4* expression in the formative state

It is intriguing that, for the high-affinity OCT4-SOX2 site substitution, we observed increased *Klf4* enhancer E2 activity by luciferase reporter assays (Fig 3C), while the corresponding endogenous mutation did not affect *Klf4* expression in the formative state (Fig 5A). This difference might indicate that enhancer E2 does not contribute *Klf4* expression, but previous work [28] and the mutants generated in this study (Fig 4B) demonstrate changes in *Klf4* expression upon endogenous manipulation of this enhancer in the naïve state.

We propose that de-activation of *Klf4* transcription when cells exit the naïve state involves more molecular mechanisms than the loss of ESRRB and STAT3 functions and, hence, loss of enhancer function. In particular, repression of *Klf4* is correlated with increased DNA methylation. While the *Klf4* enhancers are unmethylated in naïve-state mESCs, they are hypermethylated in primed-state epiblast stem cells [57]. In zebrafish, the *Klf4* promoter was hypomethylated in embryos, in which *Klf4* is expressed, but hypermethylated in fibroblasts, when *Klf4* is not expressed [58]. Thus, increased enhancer E2 activity, through improving OCT4-SOX2 binding site affinity, is likely insufficient to overcome the repressive nature of DNA methylation.

### Low-affinity OCT4-SOX2 binding sites may facilitate specificity of enhancers in different pluripotency states

While there are several previously published studies on enhancer grammar in pluripotent stem cells [21–24], it has remained unknown whether there are grammatical constraints to separate enhancer function across the different states of pluripotency. Our focus on the naïve-state specific gene *Klf4* has enabled us to gain insight into enhancer grammar that distinguishes naïve-state specific enhancers from those utilized across multiple states of pluripotency.

Our study suggests that the affinity of binding sites for OCT4 and SOX2 may delineate enhancers used in specific states of pluripotency from those general to pluripotency. Low-affinity binding sites do not sufficiently recruit OCT4 and SOX2 to enhancers to activate transcription. Thus, enhancers containing low-affinity binding sites for OCT4-SOX2 are likely combinatorial enhancers, containing clusters of heterotypic TF binding sites to enable cooperative contributions by additional TFs. We envision that the specific expression and availability of these partner TFs lead to pluripotency-state specific enhancer function. For instance, by recruiting the naïve-state TFs ESRRB and STAT3, the *Klf4* enhancers are only functional in the naïve state but not the formative and primed states (Fig 3).

By contrast, high-affinity OCT4-SOX2 binding sites can recruit OCT4 and SOX2 at sufficient levels to activate transcription without support by additional transcription factors. Hence, enhancers with high-affinity OCT4-SOX2 binding sites can function in all pluripotent states. For example, upon replacement of the native, low-affinity OCT4-SOX2 binding site with a high-affinity version, *Klf4* enhancer E2 exhibited function in both naïve-state mESCs and

formative-state EpiLCs, both in the presence and absence of the ESRRB and STAT3 binding sites (Fig 3). Additionally, enhancers for *Nanog*, *Oct4*, and *Sox2*, which are expressed in all states of pluripotency, contain relatively high-affinity binding sites for OCT4 and SOX2 (S9 Fig).

## Mechanisms for low-affinity TF binding sites in facilitating developmental gene regulation

It is well understood that suboptimal TF binding sites are key in the spatiotemporal specificity of developmental enhancers [11, 13, 40]. Many of these studies focus on developmental enhancers where the cognate transcription factors are expressed in concentration gradients across time and/or space in the embryo, leading to enhancer function only when and/or where expression of the TFs is highest [59–67]. In our study, the naïve-state specific function of *Klf4* enhancer E2 may be partially explained by reduced SOX2 expression as cells exit the naïve-state [18, 41].

However, our investigation of *Klf4* enhancer E2 reveals that suboptimal binding sites also contribute to enhancer specificity by necessitating cooperative activities from other transcription factors. As TFs cannot interact with low-affinity binding sites at a sufficient rate and/or duration to induce transcription, additional, partnering transcription factors are essential for enhancer activity. Thus, the availability of these additional transcription factors limits enhancer activities to specific cell- or tissue-types during development. Our study is, to the best of our knowledge, the first to demonstrate that low-affinity binding sites drive dependency on other transcription factors. We speculate that this mechanism may also contribute to the specific functions of previously studied low-affinity enhancers. For instance, the *t48* enhancer from *Drosophila* contains suboptimal binding sites for Dorsal and Zelda, which restrict the activities of this enhancer to the ventral region starting at nuclear cycle 14 [65]. Whereas Dorsal is expressed in a gradient along the dorsoventral axis in the fly embryo [68, 69], Zelda is uniformly expressed [70]. Hence, we envision that the low-affinity Zelda binding sites drive dependency of the *t48* enhancer on Dorsal, leading to its precise spatiotemporal expression during fly embryo development.

We acknowledge the similarity of these mechanisms, in which enhancers with low-affinity binding sites only become active when a threshold concentration of transcription factors is met. However, we propose that the presence of homotypic or heterotypic clusters of binding sites might lead to one strategy over another. For enhancers with homotypic clusters of binding sites, cell- or tissue-type specificity will be driven primarily by the concentration gradient of the cognate transcription factor. Conversely, when there are heterotypic clusters of binding sites, specificity can be achieved through the combinations of available TFs.

# Materials and methods

## Sequence analysis of OCT4-SOX2 composite binding sites

The OCT4-SOX2 composite binding sites from the *Nanog* [37, 38] and *Klf4* [28] enhancers were analyzed using the position weight matrix MA0142.1 on the JASPAR database [36].

## Mouse embryonic stem cell (mESC) culturing

JM8.N4 mESCs (obtained from the Mutant Mouse Resource & Research Centers) were grown and maintained, as previously described [28]. Briefly, mESCs were cultured feeder-free on 0.1% porcine gelatin (Sigma-Aldrich G9391) coated plates. mESC media was prepared by supplementing knockout DMEM (ThermoFisher 10829018) with 15% FBS (HyClone SH30396.03; lot number AC10235243), 1 mM GlutaMax (ThermoFisher 35050061), 0.1 mM

nonessential amino acids (ThermoFisher 11140050), 0.1 mM 2-mercaptoethanol (Sigma-Aldrich M3148), 1% penicillin-streptomycin (ThermoFisher 15140122) and 1000 units/mL LIF (Millipore ESG1107). Medium was changed every 24 hours, and mESCs were dissociated by Accutase (Sigma A6964) and passaged every one to two days, when cells reached confluency.

### Induction of epiblast-like cells (EpiLCs)

EpiLCs were induced from mESCs, as previously described [44]. 24-well or 6-well plates were pre-coated overnight with 16.7 μg/mL human plasma fibronectin (ThermoFisher PHE0023). $0.5 \times 10^5$ or $2 \times 10^5$ mESCs were seeded per well of 24-well plates or 6-well plates, respectively. mESCs were plated in EpiLC induction media [50:50 mixture of DMEM/F12 and Neurobasal media (ThermoFisher 12660012 and 21103–049), 1X N2 supplement (17502001), 1X B27 supplement (17504–044), 1 mM GlutaMax (ThermoFisher 35050061), 0.1 mM 2-mercaptoethanol (Sigma-Aldrich M3148), and 1% penicillin-streptomycin (ThermoFisher 15140122)], containing 20 ng/mL activin A (PeproTech 120-14E), 12 ng/mL bFGF (ThermoFisher PMG0031), and 1% KSR (ThermoFisher A3181502). Medium was changed every 24 hours, and inductions took a total of 48 hours.

### Construction of plasmids for luciferase reporter assays

For constructs to evaluate OCT4-SOX2 binding sites sequences, double-stranded oligonucleotides, comprising a single OCT4-SOX2 binding site sequence, were inserted at the KpnI and XhoI sites in pGL4.23 (Promega E8411). For plasmids containing the *Klf4* enhancer E2 (chr4:55475372–55476162), sequences were amplified from genomic DNA from JM8.N4 mESCs using PrimeSTAR HS DNA polymerase (Takara R044A) and inserted at the KpnI and XhoI sites in pGL4.23 (Promega E8411). To test *Klf4* enhancer E2 sequences with *Klf4* promoters, the core promoter (chr4:55532376–55532525) or core and proximal promoter (chr4:55532376–55533475) were inserted at the EcoRV and HindIII sites in pGL4.23 (Promega E6651). Enhancer E2 sequences were subsequently added to plasmids with the core promoter at KpnI and XhoI sites or to plasmids with the core and proximal promoter at KpnI and EcoRV. All constructs were confirmed by automated Sanger sequencing. All oligonucleotide sequences are provided in S1 Table.

### Mutagenesis of the *Klf4* enhancer sequences

Mutations to optimize the OCT4-SOX2 composite motifs or to disrupt the ESRRB and STAT3 binding sites in the *Klf4* enhancers were generated by overlap extension PCR. The pGL4.23 plasmids containing the wild-type *Klf4* enhancer sequences were used as the template for PCR. For each mutation, two overlapping segments of a *Klf4* enhancer were amplified by PCR, using a pair of external and internal primers. The internal primers generated the substitution mutations at the desired location and complementary 3' ends. During a subsequent PCR, the complementary ends of these overlapping segments hybridized and were extended to produce the full-length enhancer sequence for further amplification with the external primers. Sequences for primers are provided in S1 Table. Mutated *Klf4* enhancer sequences were inserted at the KpnI and XhoI sites in pGL4.23 [*luc2*/minP] (Promega E8411), and all mutations were confirmed by automated Sanger sequencing.

## Luciferase reporter gene assays

mESCs or EpiLCs were plated in 24-well plates 24 hours prior to transfection, as described above. Using Lipofectamine 3000 (ThermoFisher L3000008), cells were transfected with firefly luciferase reporter constructs (500 ng) and the CMV-*Renilla* luciferase reporter plasmid (10 ng; Promega E2261). Cells were harvested 21–24 hours post-transfection and assayed using the Dual Luciferase Reporter Assay kit, according to manufacturer protocol (Promega E1910). Luminescence from the firefly and *Renilla* luciferase proteins were measured using a Promega GloMax 20/20 Luminometer (Promega E5311). The firefly luciferase activity of each sample was normalized to the corresponding *Renilla* luciferase activity. At least three replicates were performed with cells plated on different dates. To assess statistical significance of differences, unpaired, two-sample t-tests were performed with a significance level of 0.05.

## Electrophoretic mobility shift assays (EMSAs)

Recombinant, full-length, His-tagged human OCT4 and SOX2 were expressed and purified, as previously described [28]. *Escherichia coli* BL21(DE3) cells were transformed with either the pSKB3-6H-TEV-hOCT4 or the pSKB3-6H-TEV-hSOX2 plasmids. Transformed bacteria were grown at 37°C to $A_{600} \sim 0.5$, and protein expression was induced for three hours with 1 mM IPTG. Cells were sonicated in Buffer A [20 mM Tris-HCl, pH 7.9, 500 mM NaCl, 0.2 mM PMSF, 1 mM benzamidine, 10 mM β-glycerophosphate, and EDTA-free Protease Inhibitor Cocktail (Roche)] containing 5 mM imidazole. Following centrifugation (15 min; 20,000 *g*; 4°C), the pellet was resuspended and sonicated in Buffer A, containing 5 mM imidazole and 6 M urea. The sonicated pellet was incubated at 4°C with gentle agitation and then centrifuged (15 min; 20,000 *g*; 4°C). The cleared extract was loaded onto a 1 mL His GraviTrap column (Cytiva), which had been equilibrated in Buffer A, containing 5 mM imidazole and 6 M urea. The column was subsequently washed with 25 column volumes of Buffer A containing 5 mM imidazole and 6 M urea, followed by 25 column volumes of Buffer A containing 20 mM imidazole and 6 M urea. Buffer A containing 300 mM imidazole and 6 M urea was used to elute His-tagged OCT4 or SOX2, and the denatured proteins were refolded by dialysis to Buffer A containing 2 M urea, in 1 M increments. Prior to use, purified human proteins were diluted as necessary in Dilution Buffer [25 mM K-Hepes pH 7.6, 100 mM KCl, 12.5 mM MgCl2, 0.1 mM EDTA, 0.1% NP-40, 0.5mM PMSF, 1 mM DTT, 1 mM benzamidine, and 0.2 mg/mL recombinant insulin].

Cy5-labeled DNA probes were generated as previously described [28]. The Cy5-labeled oligo (CCAGTCTCACCAAGGC) was first annealed with the template oligos containing the binding sites for OCT4 and SOX2 from the *Nanog* or *Klf4* enhancers (see S2 Table). Extension of the Cy-labeled oligo was performed with Pfu polymerase to generate a complete double-stranded DNA product, which was subsequently purified by polyacrylamide gel electrophoresis.

Binding reactions and non-denaturing gel electrophoresis were performed, as described [28]. Briefly, a typical 5 μL binding reaction contained 2 nM of Cy5-labeled probe, 19 nM purified OCT4, 16 nM purified SOX2, 1 μg/μL poly dG:dC DNA in the Binding Buffer (12.5 mM K-Hepes pH 7.6, 50 mM NaCl,1 mM MgCl2, 10% glycerol, 0.1% NP-40, 0.1 mg/mL BSA, 1 mM DTT). Reactions were incubated at 30°C for 15 minutes and analyzed by non-denaturing gel electrophoresis (4% acrylamide, 0.5X TGE). Gels were imaged on a Bio-Rad ChemiDoc XRS+ system, using the settings for Cy5 detection. Replicate EMSAs were performed at least three times on different dates, and representative images are presented in the figures.

To quantify the EMSA data, analysis was performed using the Bio-Rad Image Lab software, version 5. Using the "Lanes and Bands" function in the Analysis Tool Box, lanes were drawn

up manually to encompass the length of the gel, from just below the wells to about 1 cm below the unbound DNA band. On each gel, all lanes were set to the same width, which encompassed the entire width of the band with the most signal. Background subtraction was enabled, using a disk size of 10 mm. Percent DNA bound by OCT4-SOX2 dimers was calculated by dividing the volume intensity for the OCT4-SOX2-DNA heterotrimer band by the total volume intensity from the lane. Averages were calculated across replicate EMSA experiments, and unpaired, two-sample t-tests were performed.

## Generation of mutant mESC lines

gRNAs were designed using the Custom Alt-R CRISPR-Cas9 guide RNA tool by Integrated DNA Technologies to minimize off-targets and maximize on-target efficacy. Custom oligos, listed in S3 Table, were ordered to insert gRNA sequences at BpiI sites into a modified pX330 (Addgene #42230) vector, for co-expression of wild-type SpCas9 and a puromycin selection cassette (a kind gift from Frank Xie, Cleveland Clinic).

A homology-directed repair (HDR) template was constructed in the pBluescript II SK(+) vector. The multiple cloning site was excised and replaced with a version containing the FRT and FRT-F3 Flp recombinase target sites [71] to generate pBSII-FRT-MCS (deposited onto Addgene, #205988). The *Klf4* enhancer E2 locus (chr4:55474835–55476162, mm10) was inserted at XhoI and SacII. Primers for amplification also mutated the PAM sites, to prevent targeting by Cas9 during editing. A mCherry expression cassette under the control of the PGK promoter was inserted at NdeI and SacII. Homology arms (mm10: chr4:55476163–55476702 and chr: 55474335–55474834) were inserted at KpnI/PmlI and SmaI/SalI, respectively. All primers to amplify inserts for the HDR template are provided in S3 Table. Homology arms and insert sequences were confirmed by automated Sanger sequencing.

Plasmids for Cas9/gRNA expression and the HDR template were transfected into wild-type JM8.N4 mESCs using Lipofectamine 3000 (ThermoFisher L3000008). 24 hours after transfection, cells were FACS sorted for mCherry-positive cells, representing transfected cells. Following six additional days of growth, FACS sorting was repeated to select mCherry-positive cells. Sorted cells were plated sparsely on gelatin-coated 10 cm plates. After six days of growth, individual clones were isolated. Genomic DNA was extracted using the QuickExtract DNA Extraction Solution (Lucigen QE0905T) and analyzed by PCR, using the primers in S4 Table. Clones were verified by automated Sanger sequencing and renamed as *Klf4*-FRT-mCherry-E2-F3.

To introduce *Klf4* enhancer E2 mutant sequences into the genome, Flp recombinase was utilized to exchange sequences at the FRT and FRT-F3 sequences in the above *Klf4*-FRT-mCherry-E2-F3 mESC line. Flp recombinase was expressed from the pBSII-PGK-Flp-IRES-Puro plasmid (deposited onto Addgene #205989), which was generated using the pBluescript II SK(+) vector. The PGK promoter was inserted at the KpnI/XhoI cut sites; the optimized Flp recombinase expression cassette was amplified from pDIRE (Addgene #26745) and inserted at EcoRI/NotI; and a cassette with an internal ribosome entry sequence (IRES) sequence followed by the puromycin resistance gene was inserted at NotI/SacI.

For plasmids to exchange sequences between the FRT and FRT-F3 sites, the *Klf4* enhancer E2 locus (chr4:55474835–55476162, mm10) was amplified using primers in S3 Table and blunt-end ligated into pBSII-FRT-MCS at the NruI sites, such that the E2 sequence is between the FRT and FRT-F3 sequences. Mutant variants were generated by overlap extension PCR and confirmed by automated Sanger sequencing, as described above.

The pBSII-PGK-Flp-IRES-Puro plasmid and *Klf4* enhancer E2 exchange plasmids were co-transfected into *Klf4*-FRT-mCherry-E2-F3 mESCs using Lipofectamine 3000 (ThermoFisher L3000008) at a 1:5 mass ratio. 24 hours after transfection, transfected cells were selected by

growth in 1 μg/mL puromycin for 24 hours. After an additional 5–6 days of growth, cells were FACS sorted for mCherry-negative cells and plated sparsely for clonal outgrowth. Colonies were selected and isolated in 96-well plates, as described above, and mutations were screened by PCR and confirmed by automated Sanger sequencing. For analyses of gene-edited cells, at least six clones were assessed for wild-type enhancer E2 and each mutant enhancer E2.

## Quantitative reverse-transcription PCR

Naïve-state mESCs were grown in 6-well plates and harvested at 75–95% confluency, after two or three days of growth. Primed-state mEpiLCs were collected after two days of induction. Total RNA was extracted using the RNeasy Mini Kit (Qiagen 74104), and cDNA was synthesized with the SuperScript III reverse transcriptase (ThermoFisher 18080051). Quantitative, real-time PCR was performed on the cDNA with the SYBR Select Master Mix for CFX (ThermoFisher 4472937), using primers listed S5 Table. Three technical replicates were performed, and all data was normalized to *Gapdh*, according to the ΔΔCt method [72], to calculate a relative level of expression for each target gene. For analysis of gene-edited cells, at least six clones were assessed for wild-type or each mutant enhancer E2 sequence. A cDNA sample from non-edited mESCs or mEpiLCs was used as a reference sample for normalization across different plates. Following normalization to this standard, averages were taken across clones for wild-type or mutant enhancer E2, and the Grubb's test was used to identify outliers. Data was then normalized to the average of clones containing the wild-type enhancer E2, unless otherwise indicated. Statistical significance was assessed by calculating 95% confidence intervals and performing two-sample t-tests.

## Bright-field microscopy of mESCs

mESCs were grown on gelatinized 6-well plates until approximately 70% confluent. Cells were imaged on an Accu-Scope inverted microscope 3034, using a SPOT Idea CMOS Microscope camera, at 40X magnification. A 1 mm stage micrometer was used to determine sizing scale.

## Cell growth assays

In gelatinized 96-well plates, 10,000 naïve-state mESCs were plated per well in 100 μL of mESC media. Cells were incubated at 37˚C, with 5% $CO_2$, and medium was changed every 24 hours. At 0, 24, and 48 hours after plating cells, 10 μL of CCK-8 reagent (Selleck B34302) was added to each well, followed by a 2 h incubation at 37˚C (5% $CO_2$). Absorbance at 450 nm was measured on a BMG Labtech POLARstar Omega. Wells were scanned using a spiral average, with a 5 mm diameter. For blanks, wells containing mESC media but no cells were treated with CCK-8 reagent. Three technical replicates at each time point were performed for six different clones of wild-type and each mutant enhancer E2. The average of the blank wells on each plate was subtracted from each data point to account for background absorbance. For each clone, data was graphed as absorbance over time and fitted to an exponential curve, $y = a \times e^{bx}$, where x is time. Doubling time was calculated as ln(2)/b, where b is from the fitted exponential curve, and averaged across wild-type or mutant enhancer E2 clones. Statistical significance was assessed by performing two-sample t-tests.

## Supporting information

**S1 Fig. Timeline and key characteristics of the different states of pluripotency.** Diagram outlines how the three states of pluripotency correspond to developmental stages in mouse and human embryos. Markers and signaling pathways for each state are represented by labeled

boxes, with gradients to indicate relative levels.
(TIF)

**S2 Fig. Map of the *Klf4* enhancers and TF binding.** (upper) Schematic diagram of the *Klf4* locus, located at chromosome 4: 55,460,639–55,535,229 (mm10) in the *Mus. musculus* genome. The three *Klf4* enhancers (E1, E2, and E3) are located 55–75 kb downstream of the transcription start site and are highlighted in light red, green, and blue, respectively. (lower) Localization of OCT4, SOX2, ESRRB, and STAT3 across the *Klf4* locus. ChIP-seq or ChIP-exo data for OCT4 [73], SOX2 [74], ESRRB [28], and STAT3 [28] were uploaded onto the UCSC genome browser and viewed on the *M. musculus* mm10 genome. Black bars represent locations of peaks.
(TIF)

**S3 Fig. The OCT4-SOX2 Motifs in the *Klf4* enhancers exhibit suboptimal binding affinity for the OCT4-SOX2 complex.** (A, C, E) Electrophoretic mobility shift assays were performed with Cy5-labeled DNA probes containing the OCT4-SOX2 composite motif (OS site) from the *Klf4* enhancer E1 (A), enhancer E2 (C), or enhancer E3 (E). Binding reactions contained 2 nM of Cy5-labeled DNA probe. A dilution series of OCT4 and SOX2 was employed, in which "1X" is 19 nM OCT4 and 16 nM SOX2. A probe with the OS site from the *Nanog* enhancer was employed as a control. At least three replicate experiments were performed, and representative images are shown. (B, D, F) Quantitative analysis of EMSA data to assess OCT4-SOX2 binding to OS sites from *Klf4* enhancer E1 (B), enhancer E2 (D), and enhancer E3 (F). The % DNA bound by OCT4-SOX2 dimers (amount of signal for the DNA-OCT4-SOX2 band/total DNA signal in the lane) vs relative amount of OCT4 and SOX2 protein is shown. Error bars represent mean of replicates ± standard deviation. The gray horizontal line represents the average % DNA bound by OCT4-SOX2 dimers to the OS site from the *Nanog* enhancer.
(TIF)

**S4 Fig. Sequence of *Klf4* enhancer E2 and luciferase reporter construct with enhancer E2.** (A) Full sequence of the *Klf4* enhancer E2, with sites highlighted for the OCT4-SOX2 composite motif, ESRRB binding site, and STAT3 binding site. (B) Schematic diagram of the construct containing the *Klf4* enhancer E2 for luciferase reporter assays. Enhancer E2 was inserted into pGL4.23 at the KpnI and XhoI cut sites.
(TIF)

**S5 Fig. Presence of *Klf4* promoter elements decreases the effects of enhancer E2 in reporter assays.** The wild-type (WT) *Klf4* enhancer E2 or the high-affinity OCT4-SOX2 substitution mutant (OS+, see also Fig 3A) was cloned into a firefly luciferase reporter plasmid. The minimal promoter (minP) was subsequently replaced with the *Klf4* core promoter (-50 to +100, relative to the transcription start site) or the *Klf4* core and proximal promoters (-1000 to +100). All constructs were utilized in dual luciferase reporter assays in mESCs. Firefly luciferase reporter data was normalized to *Renilla* luciferase and then to the empty vector, lacking the *Klf4* enhancer E2 sequence (EV). Error bars represent mean of biological replicates ± 95% confidence interval.
(TIF)

**S6 Fig. Morphology of mESCs are not affected by mutations to *Klf4* enhancer E2.** mESCs with endogenous enhancer E2 mutations from Fig 4 were grown on gelatinized 6-well plates and imaged by bright-field microscopy. Each image is of a different clone for the enhancer E2 mutation, which is indicated at the top of the figure. Scale bar represents 100 μm.
(TIF)

**S7 Fig. Growth rates of mESCs are not affected by mutations to *Klf4* enhancer E2.** Assays to measure growth rates were performed on mESCs with endogenous enhancer E2 mutations from Fig 4. 10,000 cells were seeded per well in gelatinized 96-well plates, and viable cells were quantified by CCK-8 colorimetric assays at 0, 24, and 48 hours after seeding. Three technical replicates were performed for each clone at each time point. (A) Graph of absorbance over time from one wild-type enhancer E2 clone, with fitted exponential curve and calculations for doubling time. (B) Fitted exponential curves from all clones assayed. (C) Comparison of doubling rates for wild-type and mutant enhancer E2 cell lines. Error bars represent mean of clones ± 95% confidence interval. Unpaired t-tests were performed, and ns signifies not significant.
(TIF)

**S8 Fig. Changes in gene expression during induction of EpiLCs.** (A) Wild-type mESCs were converted to epiblast-like cells (EpiLCs) by growth in primed-state induction media. Samples of total RNA were collected after 48 hours of induction from EpiLCs and from mESCs, which were grown in standard mESC growth media to maintain naïve-state. Total RNA was reversed transcribed into cDNA for analysis by quantitative PCR. Data was analyzed using the ΔΔCT method. Bars represent mean ± standard deviation (*n* = 3 technical replicates). (B-G) Comparison of gene expression in EpiLCs generated from unedited, non-RMCE-generated wild-type mESCs and from mESCs containing enhancer E2 mutations from Fig 4. cDNA was synthesized from total RNA collected after 48 hours EpiLC induction, and quantitative PCR was performed to measure levels of *Klf4* (B), *Esrrb* (C), *Nanog* (D), *Sox2* (E), *Pou5f1* (F), and *Fgf5* (G). Data was analyzed using the ΔΔCT method and normalized to unedited cells. Each data point represents the average from 2–3 technical replicates for each cell line, and error bars represent mean of clones ± 95% confidence interval.
(TIF)

**S9 Fig. Predicted higher binding affinity for OCT4-SOX2 sites for genes expressed in all states of pluripotency.** Sequences for OCT4-SOX2 composite binding sites from enhancers for *Nanog*, *Oct4*, *Sox2*, and *Klf4* (enhancer E2) were obtained from published papers [28, 37, 38, 75]. The score is the relative score, as calculated on the JASPAR database, using position frequency matrix MA0142.1 [36].
(TIF)

**S1 Table. Primers for luciferase reporter assay constructs.**
(DOCX)

**S2 Table. Oligonucleotides for EMSAs.**
(DOCX)

**S3 Table. Primers for gene editing constructs.**
(DOCX)

**S4 Table. Primers for genotyping gene edited mESCs.**
(DOCX)

**S5 Table. Primers for RT-qPCR.**
(DOCX)

**S1 File. Raw quantitative data.**
(XLSX)

**S1 Raw images. Imaging and labeling information for EMSAs.**
(PDF)

## Acknowledgments

We thank Jim Kadonaga, Robert Tjian, Liangqi Xie, Claudia Cattoglio, and Grisel Cruz-Becerra for their critical reading of this manuscript. We also acknowledge Savannah Myers (Lewis & Clark College) for assistance with mutagenesis, Gina Dailey (University of California, Berkeley) for assembling the pBSII-FRT-MCS plasmid, Jeremy McWilliams and Ethan Davis (Lewis & Clark College) for assistance with R, and Liangqi Xie (Cleveland Clinic) for the Cas9-gRNA co-expression plasmid. Lastly, we thank members of the Torigoe Lab, Norma Velazquez-Ulloa, Tamily Weissman-Unni, Greg Hermann, Lindy Gewin, Greta Binford, and Yung-Pin Chen for discussions about this work.

## Author Contributions

**Conceptualization:** Jack B. Waite, RuthMabel Boytz, Sharon E. Torigoe.

**Data curation:** Sharon E. Torigoe.

**Formal analysis:** Jack B. Waite, RuthMabel Boytz, Alexis R. Traeger, Torrey M. Lind, Koya Lumbao-Conradson, Sharon E. Torigoe.

**Funding acquisition:** Sharon E. Torigoe.

**Investigation:** Jack B. Waite, RuthMabel Boytz, Alexis R. Traeger, Torrey M. Lind, Koya Lumbao-Conradson, Sharon E. Torigoe.

**Methodology:** Jack B. Waite, RuthMabel Boytz, Sharon E. Torigoe.

**Supervision:** Sharon E. Torigoe.

**Visualization:** Jack B. Waite, Sharon E. Torigoe.

**Writing – original draft:** Jack B. Waite, Sharon E. Torigoe.

**Writing – review & editing:** Jack B. Waite, RuthMabel Boytz, Torrey M. Lind, Koya Lumbao-Conradson, Sharon E. Torigoe.

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
