## [Decision Letter · Decision Letter 0]

7 May 2024

PONE-D-24-04969A suboptimal OCT4-SOX2 binding site facilitates the naïve-state specific function of a *Klf4* enhancerPLOS ONE

Dear Dr. Torigoe,

Thank you for submitting your manuscript to PLOS ONE. After careful consideration, we feel that it has merit but does not fully meet PLOS ONE’s publication criteria as it currently stands. Therefore, we invite you to submit a revised version of the manuscript that addresses the points raised during the review process.

We look forward to receiving your revised manuscript.

Kind regards,

Atsushi Asakura, Ph.D

Academic Editor

PLOS ONE

Journal Requirements:

Reviewers' comments:

Reviewer's Responses to Questions

**Comments to the Author**

1. Is the manuscript technically sound, and do the data support the conclusions?

Reviewer #1: Partly

Reviewer #2: Partly

2. Has the statistical analysis been performed appropriately and rigorously? 

Reviewer #1: Yes

Reviewer #2: I Don't Know

3. Have the authors made all data underlying the findings in their manuscript fully available?

Reviewer #1: Yes

Reviewer #2: Yes

4. Is the manuscript presented in an intelligible fashion and written in standard English?

Reviewer #1: Yes

Reviewer #2: Yes

5. Review Comments to the Author

Reviewer #1: Enhancers are well known to play pivotal roles in transcriptional regulation. In this study, the authors sought to investigate the enhancer activity of a naïve-specific gene, Klf4, in mouse ESCs. In order to assess the enhancer activities, they have performed gel shift assay, luciferase assay, and recombinase-mediated cassette exchange. As a result, in the Klf4 enhancer region, the authors identified low-affinity OCT4-SOX2 binding sites, which could explain the different transcriptional regulation between naïve and primed pluripotent states. This study is potentially interesting because it might provide new insights into the transcriptional regulation of a naïve-specific gene, Klf4. However, the authors should address the following points to ensure the accuracy of their study and strengthen their arguments before this manuscript could be considered for publication in PLOS ONE.

A Major comment

The main concern is the degree of involvement of low- or high-affinity OCT4-SOX2 binding site for Klf4 expression in mESCs. Their luciferase assay in EpiLCs showed that high-affinity OCT4-SOX2 binding site (OS+) had higher enhancer activity (Fig. 5). However, OS+ endogenous substitution in EpiLCs did not upregulate Klf4 expression (Fig. S5). What made the discrepancy? It is not clear whether the enhancer functions in the actual genome. This point should be carefully interpreted and discussed.

Other comments

1.The authors mainly investigated the enhancers for Klf4. To ensure clarity for the readers, the positional information, such as the distance between the enhancers (E1, E2, and E3) and the Klf4 TSS site, should be illustrated.

2.The authors mainly analyzed a limited number of gene expressions by qPCR in WT and mutant cells. Other phenotypes, including the colony appearance and cell proliferation potential, should be presented.

3.In Figure 4, the authors should examine expression levels of other naïve-specific genes, such as Esrrb, Tbx3, or Dppa3, or Zfp42. Also, the colony appearance of ESCs should be presented. The authors should confirm thatΔES and OS+/ΔES-cells show flattened or dome-shaped colonies.

4.In Figure 4 and S5, for the naïve to formative transition experiment in enhancer-substituted cells, the authors should compare naïve and pluripotent gene expression before and after the transition.

5.Previous studies showed that Klf family members, including Klf2, 4, and 5, work in concert with other family members to facilitate self-renewal or maintain the undifferentiated state (Nat Cell Biol. 2008;10(3):353-60, Nat Commun. 2014;5:3719). How about the expression of other Klf family members in enhancer-substituted cells? Also, is there any difference in their proliferation potential?

Reviewer #2: Torigoe and colleagues use the composite SoxOct element to study the DNA motif grammar associated with pluripotency using a specific example. They focus on the auto-regulation of pluripotency genes and examine how Sox/Oct4 drive Klf4 expression in pluripotent states in vitro using putative Sox2/Oct4 binding sites. This follows on an GnD paper from 2017 where the same enhancer elements were studied. They perform luciferase assays, EMSAs and studies in mouse ESCs in pre- and post-implantation culture conditions. EMSAs and luciferase assays suggest that cooperative binding and gene activation is weaker in the three KLF4 enhancer. They then further analyse site E2 that this critical for Klf4 activation despite the reduced dimer formation using luciferase assays and RCME. The main takeaway of the study is that degenerate binding sites are important for context-dependent gene regulation. Whilst this is not new per se it’s interesting to show this for this enhancer element. The advance of the study is incremental but of interest to those with a focus on gene regulation in ESCs and the role of different cis-regulatory elements.

General comments:

• The introduction should elaborate on alternative SoxOct binding sites such as the compressed motif targeted by SOX17/OCT4 and the FGF4 SoxOct element with 3bp spacer

• EMSAs in Figure 2, and elsewhere are of good quality and could be used to calculate the cooperativity factor omega (all four microstates would need to be quantified using densitometry). The study would benefit from a quantitative analysis of the binding affinity or cooperativity (authors use the term affinity a lot without quantitation). Related ‘sub-optimal’ motifs might have been studied elsewhere and may help interpretation (i.e. https://doi.org/10.1093/nar/gkw1198). The overall claim that cooperative binding is Nanog>E1>E2>E3 looks convincing. Ideally, all experiments should be performed in triplicates, monomer Sox and Oct lanes should be included and quantitation provided. I realize that the Sox2 band is a bit weak in most lanes where two proteins are present. If estimations of omega are not possible for the weak Sox lane other quantitative comparisons could be considered (i.e. plotting the dimer fraction for the various DNA probes normalized by a control). In methods, please provide more details for EMSAs. Rather than the quantities of proteins provide molar concentrations for all reactants. Are these full-length proteins expressed in E.coli? Authors refer to ref28 but key points should be reported here.

• Suggest to show a UCSC ChIP-seq signal genome browser track for E1, E2, E3. This could easily be obtained from public ChIP-seq data sets (i.e. https://compbio-zhanglab.org/CRCistrome/index.php). I realise some of this is shown in ref28.

• Data in Figure 3 are interesting but could be explained more clearly in the results. Please show the sequence and indicate the length of the reporter construct in Figure 3. Here and elsewhere make clear which cells were analysed (same in Figure 5).

• Were the RCME-based reporter assays performed in both naïve ESCs and EpiLCs? Suggest showing EpiLC and naïve reporter data for the enhancer mutants side by side to back up the key claim that the motif degeneracy is the key for the specific expression of Klf4 in naïve ESCs.

• Are the enhancers E1-E3 conserved in humans?

6. PLOS authors have the option to publish the peer review history of their article (what does this mean?). If published, this will include your full peer review and any attached files.

Reviewer #1: No

Reviewer #2: No

---

## [Author Response · Author response to Decision Letter 0]

17 Jun 2024

Reviewer #1: Enhancers are well known to play pivotal roles in transcriptional regulation. In this study, the authors sought to investigate the enhancer activity of a naïve-specific gene, Klf4, in mouse ESCs. In order to assess the enhancer activities, they have performed gel shift assay, luciferase assay, and recombinase-mediated cassette exchange. As a result, in the Klf4 enhancer region, the authors identified low-affinity OCT4-SOX2 binding sites, which could explain the different transcriptional regulation between naïve and primed pluripotent states. This study is potentially interesting because it might provide new insights into the transcriptional regulation of a naïve-specific gene, Klf4. However, the authors should address the following points to ensure the accuracy of their study and strengthen their arguments before this manuscript could be considered for publication in PLOS ONE.

A Major comment

The main concern is the degree of involvement of low- or high-affinity OCT4-SOX2 binding site for Klf4 expression in mESCs. Their luciferase assay in EpiLCs showed that high-affinity OCT4-SOX2 binding site (OS+) had higher enhancer activity (Fig. 5). However, OS+ endogenous substitution in EpiLCs did not upregulate Klf4 expression (Fig. S5). What made the discrepancy? It is not clear whether the enhancer functions in the actual genome. This point should be carefully interpreted and discussed.

We thank Reviewer #1 for this question and suggestion. In the manuscript text (see page 18, lines 380-393), we have now expanded our discussion of our data to address questions about the discrepancy between the luciferase reporter assays and the mutant EpiLCs.

Other comments

1.The authors mainly investigated the enhancers for Klf4. To ensure clarity for the readers, the positional information, such as the distance between the enhancers (E1, E2, and E3) and the Klf4 TSS site, should be illustrated.

We agree with Reviewer #1 that this would be informative and helpful to readers. The NEW S2 Figure includes a diagram to map the enhancers and TSS for Klf4.

2.The authors mainly analyzed a limited number of gene expressions by qPCR in WT and mutant cells. Other phenotypes, including the colony appearance and cell proliferation potential, should be presented.

We thank Reviewer #1 for these suggestions, as further analysis of our wild-type and mutant cell lines would strengthen our study. We have now performed more qPCR measurements (REVISED Figures 4 and 5), included images of colony morphology (NEW S6 Figure), and added growth assay data assessing cell proliferation potential (NEW S7 Figure). Specifics for these additional experiments are described below, in responses to other points by Reviewer #1.

3.In Figure 4, the authors should examine expression levels of other naïve-specific genes, such as Esrrb, Tbx3, or Dppa3, or Zfp42. Also, the colony appearance of ESCs should be presented. The authors should confirm thatΔES and OS+/ΔES-cells show flattened or dome-shaped colonies.

We have addressed this comment by performing new qPCR analyses to measure levels of the naïve-state specific genes Esrrb and Tbx3 for our wild-type and mutant cell-lines in the naïve-state. These data are included in the REVISED Figure 4. We also have characterized and included other phenotypes. We have added images for colony appearance for our wild-type and mutant cell lines, which are shown in NEW S6 Figure.

4.In Figure 4 and S5, for the naïve to formative transition experiment in enhancer-substituted cells, the authors should compare naïve and pluripotent gene expression before and after the transition.

To address this important point, we have now included additional, new qPCR analyses for Pou5f1, Sox2, Nanog, Esrrb, and Tbx3 for our wild-type and mutant cell-lines in the induced formative-state EpiLCs. We also have added qPCR results for these cell-lines before induction. These are included in the REVISED Figure 5 (Figure S5 in the original submission).

5.Previous studies showed that Klf family members, including Klf2, 4, and 5, work in concert with other family members to facilitate self-renewal or maintain the undifferentiated state (Nat Cell Biol. 2008;10(3):353-60, Nat Commun. 2014;5:3719). How about the expression of other Klf family members in enhancer-substituted cells? Also, is there any difference in their proliferation potential?

While KLF2, KLF4, and KLF5 form a circuit to promote naïve-state pluripotency and self-renewal, prior studies have not indicated that they regulate each other. However, Klf2 and Klf5 appear to be naïve-state specific. So, we have performed new qPCR analyses to measure levels of Klf2 and Klf5 in our mutant cell-lines. These are included in the REVISED Figure 4.

We have also performed new cell growth assays to assess cell proliferation potential on our mutant cell lines. These data are presented in the NEW S7 Figure.

Reviewer #2: Torigoe and colleagues use the composite SoxOct element to study the DNA motif grammar associated with pluripotency using a specific example. They focus on the auto-regulation of pluripotency genes and examine how Sox/Oct4 drive Klf4 expression in pluripotent states in vitro using putative Sox2/Oct4 binding sites. This follows on an GnD paper from 2017 where the same enhancer elements were studied. They perform luciferase assays, EMSAs and studies in mouse ESCs in pre- and post-implantation culture conditions. EMSAs and luciferase assays suggest that cooperative binding and gene activation is weaker in the three KLF4 enhancer. They then further analyse site E2 that this critical for Klf4 activation despite the reduced dimer formation using luciferase assays and RCME. The main takeaway of the study is that degenerate binding sites are important for context-dependent gene regulation. Whilst this is not new per se it’s interesting to show this for this enhancer element. The advance of the study is incremental but of interest to those with a focus on gene regulation in ESCs and the role of different cis-regulatory elements.

General comments:

• The introduction should elaborate on alternative SoxOct binding sites such as the compressed motif targeted by SOX17/OCT4 and the FGF4 SoxOct element with 3bp spacer

We thank Reviewer #2 for bringing this to our attention. We have added discussion of alternative oct-sox motifs into the beginning of Results (see pages 4 and 5, lines 77-88), when we initially discuss our observations about the oct-sox motifs in the Klf4 enhancers.

• EMSAs in Figure 2, and elsewhere are of good quality and could be used to calculate the cooperativity factor omega (all four microstates would need to be quantified using densitometry). The study would benefit from a quantitative analysis of the binding affinity or cooperativity (authors use the term affinity a lot without quantitation). Related ‘sub-optimal’ motifs might have been studied elsewhere and may help interpretation (i.e. https://doi.org/10.1093/nar/gkw1198). The overall claim that cooperative binding is Nanog>E1>E2>E3 looks convincing. Ideally, all experiments should be performed in triplicates, monomer Sox and Oct lanes should be included and quantitation provided. I realize that the Sox2 band is a bit weak in most lanes where two proteins are present. If estimations of omega are not possible for the weak Sox lane other quantitative comparisons could be considered (i.e. plotting the dimer fraction for the various DNA probes normalized by a control). In methods, please provide more details for EMSAs. Rather than the quantities of proteins provide molar concentrations for all reactants. Are these full-length proteins expressed in E.coli? Authors refer to ref28 but key points should be reported here.

We thank Reviewer #2 for their comments and suggestions on our EMSA experiments. Reduced cooperative binding by OCT4 and SOX2 at the Klf4 enhancers is one potential interpretation of our data, requiring calculation of the cooperativity factor omega for each sequence. Unfortunately, as also noted by Reviewer #2, the SOX2-DNA band being undetectable in most cases, so we are unable to confidently quantify monomer binding and, hence, calculate cooperativity factor omega.

Nevertheless, we agree that quantification of our EMSA results would strengthen our conclusions about differences in binding affinity of the OCT4-SOX2 dimer to our DNA probes. We have calculated the percentage of DNA bound by the OCT4-SOX2 dimer, which is normalized to the overall Cy5 signal in each lane. These data have been added to REVISED Figure 2 and S3 Figure (formerly S2 Figure).

We also thank Reviewer #2 for their reference to the Chang, et al, 2017, Nucleic Acids Research paper. In that study, the authors examined the effects of spacing, orientation and some sequence alterations on cooperative binding by OCT and SOX proteins, including OCT4 and SOX2. While their sequence analyses do not include the differences we observe in the OCT4-SOX2 sites from the Klf4 enhancers, this study emphasizes that OCT4 and SOX2 cooperative binding is significantly impacted by the spacing and relative orientations of the individual OCT4 and SOX2 binding sites. Based on this study, the OCT4-SOX2 sites in the Klf4 enhancers have the optimal spacing and relative orientation. We have revised our text to discuss this study and its relevance to the OCT4-SOX2 sites from the Klf4 enhancers (see pages 4 and 5, lines 80-83).

We also would like to note that, as requested by Reviewer #2, we had done our EMSAs at least in triplicate prior to the original submission, as noted in the figure legends. We apologize if this information was not made more clear in the original submission. We have added a note about replicates into the Methods for EMSAs (see page 26, lines 560-561).

Lastly, we agree that more information about the EMSAs would be useful. Our revised Methods now contain more complete information about our protocols (see pages 24-26; lines 529-570). Where appropriate, we also converted to molar concentrations, namely for OCT4, SOX2 and the Cy5-labeled probes. For other reagents, concentrations are provided in units according to common practice.

• Suggest to show a UCSC ChIP-seq signal genome browser track for E1, E2, E3. This could easily be obtained from public ChIP-seq data sets (i.e. https://compbio-zhanglab.org/CRCistrome/index.php). I realise some of this is shown in ref28.

We agree with Reviewer #2 that showing genome browser tracks from ChIP-seq data would reiterate that OCT4, SOX2, ESRRB, and STAT3 bind to the Klf4 enhancers. We have generated NEW S2 Figure with genome browser tracks for all four of these transcription factors.

• Data in Figure 3 are interesting but could be explained more clearly in the results. Please show the sequence and indicate the length of the reporter construct in Figure 3. Here and elsewhere make clear which cells were analysed (same in Figure 5).

We have revised the text, in which we discuss Figure 3 (see pages 10 and 11, lines 196-219), and we hope that the revised explanations are more clear for readers.

We agree that it would be helpful for readers to have further information about our mutant enhancer E2 sequences and the reporter constructs. We have made the following changes to our manuscript:

• We have added the chromosome coordinates for enhancer E2 into our Methods, similar to other genomic DNA coordinates that we also provided. We apologize for their omission in the original submission, as those would be useful to readers.

• We have generated NEW S4 Figure, which includes (A) the entire Klf4 enhancer E2 sequence used in the luciferase reporter assays, with relevant TF binding sites labeled and (B) a schematic map of the constructs for luciferase reporter assays with Klf4 enhancer E2.

• The REVISED Figure 3A now displays sequences for the relevant TF binding sites, as well as the mutations we utilized in our study.

We hope that these revisions address the questions from Reviewer #2 about the sequences and constructs used in Figure 3.

To all figures with data from mESCs or EpiLCs, we have now added notations onto the figures to more clearly designate which cells were being analyzed.

• Were the RCME-based reporter assays performed in both naïve ESCs and EpiLCs? Suggest showing EpiLC and naïve reporter data for the enhancer mutants side by side to back up the key claim that the motif degeneracy is the key for the specific expression of Klf4 in naïve ESCs.

All luciferase reporter assays and qPCR analyses of RMCE-generated cell lines were performed in both mESCs and EpiLCs. As noted with the previous comment, we have now added notations onto figures to clearly indicate the cell-type utilized in each experiment.

We thank Reviewer #2 for the suggestion to show naïve-state and primed-state data side-by-side, as this will more effectively demonstrate the naïve-state specific function of Klf4 enhancer E2. We have re-configured Figures 3 and 5 into a single figure, putting all the luciferase reporter data together into REVISED Figure 3. In the REVISED Figure 5 (formerly S5 Figure), qPCR analyses of our RMCE-generated cell-lines from both before and after induction to EpiLCs are shown side-by-side.

• Are the enhancers E1-E3 conserved in humans?

We thank Reviewer #2 for this excellent question, as we also have wondered about the conservation of the Klf4 enhancers across species, particularly in humans. We have utilized data from ENCODE to identify candidate enhancers for Klf4 in humans. We found one possible sequence in a similar location as E1 and E2 (about 50 kb downstream of the TSS) that had predicted binding sites for OCT4, SOX2, ESRRB, and STAT3. Analysis of OCT4 ChIP-seq data from human ESCs revealed a peak at this location, and the OCT4-SOX2 composite site appears to be suboptimal, based on initial scans with JASPAR. While promising, further experimentation is required to confirm functionality of this enhancer in human ESCs, which is beyond the scope of the current manuscript.

---

## [Decision Letter · Decision Letter 1]

3 Sep 2024

PONE-D-24-04969R1A suboptimal OCT4-SOX2 binding site facilitates the naïve-state specific function of a *Klf4* enhancerPLOS ONE

Dear Dr. Torigoe,

Thank you for submitting your manuscript to PLOS ONE. After careful consideration, we feel that it has merit but does not fully meet PLOS ONE’s publication criteria as it currently stands. Therefore, we invite you to submit a revised version of the manuscript that addresses the points raised during the review process.

We look forward to receiving your revised manuscript.

Kind regards,

Miquel Vall-llosera Camps

Senior Staff Editor

PLOS ONE

**Journal Requirements:**

Reviewers' comments:

Reviewer's Responses to Questions

**Comments to the Author**

1. If the authors have adequately addressed your comments raised in a previous round of review and you feel that this manuscript is now acceptable for publication, you may indicate that here to bypass the “Comments to the Author” section, enter your conflict of interest statement in the “Confidential to Editor” section, and submit your "Accept" recommendation.

Reviewer #1: (No Response)

Reviewer #2: All comments have been addressed

2. Is the manuscript technically sound, and do the data support the conclusions?

Reviewer #1: Yes

Reviewer #2: Yes

3. Has the statistical analysis been performed appropriately and rigorously? 

Reviewer #1: Yes

Reviewer #2: I Don't Know

4. Have the authors made all data underlying the findings in their manuscript fully available?

Reviewer #1: Yes

Reviewer #2: Yes

5. Is the manuscript presented in an intelligible fashion and written in standard English?

Reviewer #1: Yes

Reviewer #2: Yes

6. Review Comments to the Author

**Reviewer #1:** Although the impact of low- and high-affinity Oct4-Sox2 binding site on the expression of Klf4 or the functional regulation of pluripotency, such as other naïve-related gene expressions and colony appearance, is not highly significant, their additional experiments clarified the claims in the manuscript. As the authors discussed, further studies will be required to understand the mechanisms involved in the enhancer activity. Their revised manuscript is now worth considering for publication in PLOS One. My additional minor comments on the manuscript are as follows:

Minor comments:

1. Line 376-393: These discussions should be included in the Discussion section.

2. Line 377 and 380: Fig 5B -> Fig 5A

**Reviewer #2: **The authors have addressed all my comments and I am supportive of accepting this nice work. I'd encourage authors to ensure that statistical tests and replications of experiments follow rigorous standards and that plasmids and raw data are made available in public repositories (figshare, add gene) if appropriate.

7. PLOS authors have the option to publish the peer review history of their article (what does this mean?). If published, this will include your full peer review and any attached files.

Reviewer #1: No

Reviewer #2: No

---

## [Author Response · Author response to Decision Letter 1]

6 Sep 2024

Reviewer #1: Although the impact of low- and high-affinity Oct4-Sox2 binding site on the expression of Klf4 or the functional regulation of pluripotency, such as other naïve-related gene expressions and colony appearance, is not highly significant, their additional experiments clarified the claims in the manuscript. As the authors discussed, further studies will be required to understand the mechanisms involved in the enhancer activity. Their revised manuscript is now worth considering for publication in PLOS One.

We thank Reviewer #1 for their review of our revised manuscript. We appreciate that they find our additional experiments clarifying of our conclusions, such that the manuscript meets expectations for publication.

My additional minor comments on the manuscript are as follows:

Minor comments:

1. Line 376-393: These discussions should be included in the Discussion section.

We have moved these to the Discussion section and have edited the text accordingly.

2. Line 377 and 380: Fig 5B -> Fig 5A

We apologize for the typos in reference to Figure 5. We have corrected these in the manuscript.

Reviewer #2: The authors have addressed all my comments and I am supportive of accepting this nice work. I'd encourage authors to ensure that statistical tests and replications of experiments follow rigorous standards and that plasmids and raw data are made available in public repositories (figshare, add gene) if appropriate.

We thank Reviewer #2 for their review of our revised manuscript, and we are glad to hear that we have addressed all of their comments. We appreciate their recommendation to accept our manuscript for publication.

We have made a few minor edits to the Methods section to add more details to clarify our statistical tests and replications of experiments. Some of these details were included in the figure legends, and they are now added to the Methods.

Plasmids have already been deposited to Addgene, and they will be made available when the manuscript is published. All raw quantitative data was included with the revised manuscript as a Supporting Information file (S1_file) to be made available with the paper upon publication. Raw images for EMSAs are already available on the Open Science Framework, and DOI information was provided in the submission information for the revised manuscript.

---

## [Editor Report · Decision Letter 2]

13 Sep 2024

A suboptimal OCT4-SOX2 binding site facilitates the naïve-state specific function of a *Klf4* enhancer

PONE-D-24-04969R2

Dear Dr. Torigoe,

We’re pleased to inform you that your manuscript has been judged scientifically suitable for publication and will be formally accepted for publication once it meets all outstanding technical requirements.

Kind regards,

Miquel Vall-llosera Camps

Senior Staff Editor

PLOS ONE
---

## [Editor Report · Acceptance letter]

18 Sep 2024

PONE-D-24-04969R2 

PLOS ONE

Dear Dr. Torigoe, 

I'm pleased to inform you that your manuscript has been deemed suitable for publication in PLOS ONE. Congratulations! Your manuscript is now being handed over to our production team.

Kind regards, 

on behalf of

Dr. Miquel Vall-llosera Camps 

Staff Editor

PLOS ONE